# Variability in the use of pulse oximeters with children in Kenyan hospitals: A mixed-methods analysis

Abigail J. Enoch[1]*, Mike English[2,3], the Clinical Information Network[¶], Gerald McGivern[4], Sasha Shepperd[5]*

**1** Nuffield Department of Population Health, University of Oxford, Oxford, United Kingdom (former DPhil student), **2** KEMRI-Wellcome Trust Research Programme, Nairobi, Kenya, **3** Nuffield Department of Medicine, University of Oxford, Oxford, United Kingdom, **4** Warwick Business School, University of Warwick, Coventry, United Kingdom, **5** Nuffield Department of Population Health, University of Oxford, Oxford, United Kingdom

[¶]Membership of the Clinical Information Network is provided in the Acknowledgments
* abby.enoch@gmail.com (AE); sasha.shepperd@ndph.ox.ac.uk (SS)

**Data Availability Statement:** Quantitative data: Patient level data used in this report are under the primary jurisdiction of the Ministry of Health in Kenya. The authors had full access to all the

## Abstract

### Background

Pulse oximetry, a relatively inexpensive technology, has the potential to improve health outcomes by reducing incorrect diagnoses and supporting appropriate treatment decisions. There is evidence that in low- and middle-income countries, even when available, widespread uptake of pulse oximeters has not occurred, and little research has examined why. We sought to determine when and with which children pulse oximeters are used in Kenyan hospitals, how pulse oximeter use impacts treatment provision, and the barriers to pulse oximeter use.

### Methods and findings

We analyzed admissions data recorded through Kenya's Clinical Information Network (CIN) between September 2013 and February 2016. We carried out multiple imputation and generated multivariable regression models in R. We also conducted interviews with 30 healthcare workers and staff from 14 Kenyan hospitals to examine pulse oximetry adoption. We adapted the Integrative Model of Behavioural Prediction to link the results from the multivariable regression analyses to the qualitative findings. We included 27,906 child admissions from 7 hospitals in the quantitative analyses. The median age of the children was 1 year, and 55% were male. Three-quarters had a fever, over half had a cough; other symptoms/signs were difficulty breathing (34%), difficulty feeding (34%), and indrawing (32%). The most common diagnoses were pneumonia, diarrhea, and malaria: 45%, 35%, and 28% of children, respectively, had these diagnoses. Half of the children obtained a pulse oximeter reading, and of these, 10% had an oxygen saturation level below 90%. Children were more likely to receive a pulse oximeter reading if they were not alert (odds ratio [OR]: 1.30, 95% confidence interval (CI): 1.09, 1.55, $p$ = 0.003), had chest indrawing (OR: 1.28, 95% CI: 1.17, 1.40, $p$ < 0.001), or a very high respiratory rate (OR: 1.27, 95% CI: 1.13, 1.43, $p$ <

quantitative data in the study and take responsibility for the integrity of the data and the accuracy of the data analysis. Researchers can contact the KEMRI Wellcome Trust Research Programme Data Governance Committee (dgc@kemri-wellcome.org) for advice on accessing the data including whether a new ethical review submission and approval from the Ministry of Health in Kenya would be required. Qualitative data: The interview data cannot be open access as this would not comply with the consent process: the interviewees were informed that only the researchers would have access to the transcripts and that the transcripts would be destroyed after a specified amount of time following the end of the study. In addition, we are concerned that interview data differ to de-identified quantitative data as it might be possible to identify individuals from the full transcript.

**Funding:** AJE is the recipient of a studentship jointly funded by the Medical Research Council and the Oxford Nuffield Department of Population Health; ME is supported by funds from a Wellcome Trust Senior Fellowship (#207522). SS is supported by Oxford's Nuffield Department of Population Health. The funders had no role in study design, data collection and analysis, decision to publish, or preparation of the manuscript.

**Competing interests:** I have read the journal's policy and the authors of this manuscript have the following competing interests: ME receives research funding from the Wellcome Trust and the Gates Foundation to study implementation in Kenya and this includes ongoing work that studies pulse oximetry amongst other technologies.

**Abbreviations:** CI, confidence interval; CIN, Clinical Information Network; FCS, fully conditional specification; HCW, healthcare worker; KEMRI, Kenya Medical Research Institute; LMIC, low- and middle-income country; MAR, Missing At Random; MICE, Multivariate Imputation by Chained Equations; OR, odds ratio; PAR, Paediatric Admission Record; SERU, KEMRI's Scientific and Ethics Review Unit; WHO, World Health Organization.

0.001), as were children admitted to certain hospitals, at later time periods, and when a Paediatric Admission Record (PAR) was used (OR PAR used compared with PAR not present: 2.41, 95% CI: 1.98, 2.94, $p < 0.001$). Children were more likely to be prescribed oxygen if a pulse oximeter reading was obtained (OR: 1.42, 95% CI:1.25, 1.62, $p < 0.001$) and if this reading was below 90% (OR: 3.29, 95% CI: 2.82, 3.84, $p < 0.001$). The interviews indicated that the main barriers to pulse oximeter use are inadequate supply, broken pulse oximeters, and insufficient training on how, when, and why to use pulse oximeters and interpret their results. According to the interviews, variation in pulse oximeter use between hospitals is because of differences in pulse oximeter availability and the leadership of senior doctors in advocating for pulse oximeter use, whereas variation within hospitals over time is due to repair delays. Pulse oximeter use increased over time, likely because of the CIN's feedback to hospitals. When pulse oximeters are used, they are sometimes used incorrectly and some healthcare workers lack confidence in readings that contradict clinical signs. The main limitations of the study are that children with high levels of missing data were not excluded, interview participants might not have been representative, and the interviews did not enable a detailed exploration of differences between counties or across senior management groups.

## Conclusions

There remain major challenges to implementing pulse oximetry—a cheap, decades old technology—into routine care in Kenya. Implementation requires efficient and transparent procurement and repair systems to ensure adequate availability. Periodic training, structured clinical records that include prompts, the promotion of pulse oximetry by senior doctors, and monitoring and feedback might also support pulse oximeter use. Our findings can inform strategies to support the use of pulse oximeters to guide prompt and effective treatment, in line with the Sustainable Development Goals. Without effective implementation, the potential benefits of pulse oximeters and possible hospital cost-savings by targeting oxygen therapy might not be realized.

## Author summary

### Why was this study done?

- Pulse oximeters are an easy to use, relatively inexpensive technology that helps to detect low levels of oxygen in the blood, which assists healthcare workers in determining a child's diagnosis and appropriate treatment.

- Studies show that, even when available, pulse oximeters are often not used in low- and middle-income countries, but little research has looked into why.

- We therefore carried out this study to determine when and why healthcare workers do or do not use pulse oximeters with children admitted to Kenyan hospitals.

**What did the researchers do and find?**

- We carried out statistical analyses on a data set of 27,906 children admitted to 7 Kenyan hospitals and interviews with 30 healthcare workers and staff at 14 Kenyan hospitals.

- We found that there was variability in the use of pulse oximeters and that healthcare workers were most likely to use pulse oximeters with children in certain hospitals, at later time periods, and with children who were not alert or had chest indrawing or a high respiratory rate.

- The main factors that prevent healthcare workers from using pulse oximeters appropriately are if there is an inadequate supply, a delay in repairing broken pulse oximeters, and the healthcare workers have not had sufficient training on when, how, and why to use pulse oximeters and interpret their results.

**What do these findings mean?**

- The findings suggest that healthcare workers are likely to use pulse oximeters with more children if there are efficient and transparent systems for procurement and repair, oxygen therapy is available, training and feedback are provided, and senior doctors advocate for the use of pulse oximeters.

- If healthcare workers use pulse oximeters with more children at admission, this may increase the number of children who are correctly diagnosed and appropriately treated, potentially leading to fewer child deaths.

## Introduction

Pulse oximeters are an easy to use, effective, low-cost technology intended to improve health outcomes by detecting hypoxemia (low blood oxygen levels) [1]. Evidence suggests pulse oximeters identify 20% to 30% additional hypoxic children compared with using clinical signs alone, e.g., grunting and depressed consciousness, which can be imprecise [1,2]. Hypoxemia is associated with increased risk of death in children and adolescents and is a common complication of pneumonia (the leading infectious cause of death in children aged 1 month to 5 years worldwide), bronchiolitis, asthma, and other serious conditions (e.g., sepsis) [1,3–6].

Recognizing the potential to improve health outcomes by reducing the number of incorrect diagnoses and supporting treatment decisions, there are a number of international initiatives aimed at increasing pulse oximeter availability [7–13]. However, evidence suggests that even when pulse oximeters are available they are often not used [14,15]. Little research in any setting has examined why. The limited evidence suggests healthcare workers' (HCWs) decision to use pulse oximeters may vary depending on children's characteristics, such as age, respiratory symptoms, diagnosis, and admission date, and that barriers to pulse oximeter use may include insufficient training, inadequate guidelines, and repair issues [16–22].

Understanding and addressing the barriers to using pulse oximeters in a low- and middle-income country (LMIC) such as Kenya has the potential to help reduce mortality in children needing oxygen, because widespread dissemination and uptake of pulse oximetry has not

occurred [23]. As part of a strategy to improve the quality of pediatric care in Kenya, a Clinical Information Network (CIN) was established in 2013 to capture standardized clinical data on hospital admissions [24–26]. Fourteen hospitals across 12 of the 47 counties in Kenya are part of this CIN. We analyzed these data, combined with interviews, to examine pulse oximetry adoption in Kenyan hospitals using a mixed-methods approach.

We sought to determine when and with which groups of children pulse oximeters are used in Kenyan hospitals, how pulse oximeter use impacts treatment provision, and the barriers to pulse oximeter use. Our aim was to inform the development of effective strategies to overcome barriers, promote pulse oximeter use, improve treatment, and therefore reduce child mortality rates.

## Methods

### Setting

Kenya has a population of over 48 million, and, in 2016, its total health expenditure was US$66 per capita, which corresponded to 4.6% of GDP [27,28]. In 2015, 75,000 children under the age of 5 died, which corresponded to an under-5 mortality rate of 51 per 1,000 live births [29]. In 2013, there was 1 medical doctor and 12 nurses per 10,000 population, compared with the minimum threshold of 23 HCWs set by the WHO [30,31]. Furthermore, HCW distribution is very uneven across the country [31,32]. Local guidelines on care for children at admission to Kenyan hospitals, The Kenyan Basic Paediatric Protocols, have recommended pulse oximetry since 2013 [33], and a widely used training program (ETAT+ [34]) supports routine measurement of oxygen saturation in all admitted children.

### Ethical approval

We obtained ethical approval for analyzing CIN data and for conducting interviews from the University of Oxford's CUREC and OXTREC ethical committees, and from the Kenya Medical Research Institute's (KEMRI's) Scientific and Ethics Review Unit (SERU). We also obtained permission from the Kenyan hospitals' medical superintendent and pediatrician to visit the hospitals and interview staff, who provided informed consent for participation and the audio recording of their interviews.

### Mixed-methods design

We used a sequential mixed-methods approach [35] to investigate the use of pulse oximeters during the pediatric admission assessment. We used CIN data over 2.5 years to examine patterns of pulse oximeter use and associations with patients' characteristics. We used qualitative data to examine the findings of the quantitative analyses and to explore why and how HCWs use or do not use pulse oximeters.

### The CIN

Data were collected by trained clerks from the paper medical records of children admitted to 14 public (district) hospitals that are first referral centers. Core data domains were demographics, symptoms, diagnoses, treatments, and outcomes. Data were entered into an electronic system and de-identified, and data quality was checked electronically and manually; this included feedback on completeness and adherence to care recommendations through audit reports every 2 to 3 months. The process of data collection and management is described in full elsewhere [25,26,36].

## Data management and analysis

We analyzed data collected from 7 of the CIN hospitals, selected because pulse oximeters were used on average with at least 20% of the children admitted between September 2013 and February 2016. Hospitals with lower rates were excluded to ensure adequate data on pulse oximeter measure events were available for analysis.

We included data from admissions from September 2013 (the start of CIN data collection) through February 2016. We excluded children under 1 month, because they have different health issues and treatments compared with older children [37], and patients over 12 years. Our analyses focused on the admission assessment and did not capture data on pulse oximeter use later in the course of admission. After cleaning and organizing the data, we used 45 variables describing children's demographics, symptoms, diagnostic tools used, diagnoses, treatments, and outcome.

Each of the variables had less than 25% missing data (most had less than 10%), and each admission episode had a median of 1 missing value (mean = 3.2). We explored missing data patterns to determine whether the data could be treated as Missing at Random (MAR) by examining whether specific variables and values influenced the likelihood of missing data (see S1 Text). To enable use of all the available data, we carried out multiple imputation (30 imputations), using a fully conditional specification (FCS) Multivariate Imputation by Chained Equations (MICE) approach in R [38].

We used the "mice" program in R to generate multivariable logistic regression models using backwards stepwise regression. A model was created to examine factors associated with pulse oximeter use and another, using a similar set of variables, for factors associated with oxygen provision. We assessed the robustness of this method by comparing the variables, odds ratios (ORs), and confidence intervals (CIs) with those from more complex models that used bootstrapping. (See S2 Text for more details on this process and selection of variables).

The methods described above closely followed the protocol (see S1 Study Protocol), with the exception that (i) we limited the number of variables included in the analysis following a further review of the evidence and advice from topic experts; (ii) we originally planned to analyze 2 subgroups, children aged between 1 month and 59 months and those aged 1 month to 12 years, but it became clear that the number of admissions aged 5 and over were low, and CIN clinicians agreed that admission assessment guidelines for the under-5 population are uniformly applied to older age groups; (iii) we originally planned to analyze pulse oximetry data by the thresholds <85%, 85% to 89%, 90% to 94%, and >95%, but numbers in the lower groups were small, and we judged that grouping high versus very high readings would probably not be useful, so we limited the comparison to <90% versus ≥90%; (iv) and we originally planned to assess the influence of pulse oximeter use on oxygen therapy and antibiotic prescribing but later decided to just focus on oxygen therapy.

## Qualitative methods

We collected qualitative data from all 14 CIN hospitals. We conducted direct (nonparticipant) observations of the delivery of healthcare in 3 hospitals to obtain an understanding of the pediatric admission process and to inform the content of the topic guide; these hospitals were selected to observe settings with different levels of pulse oximetry adoption. This was followed by 16 semistructured audio-recorded and transcribed interviews with HCWs (pediatricians, medical officers/interns, clinical officers/interns, and nurses) from 4 hospitals; shorter interviews with 2 procurement officials and one medical superintendent from 2 of these 4 hospitals and, at a network meeting, with 11 HCWs from 10 other hospitals. Both types of interviews

covered similar topics. The shorter interviews were less formal, lasted up to 15 minutes versus 20 to 35 minutes, and were recorded through field notes rather than audio recordings. Both sets of interviews contributed to the findings; quotes were taken from the longer audio-recorded interviews. These will henceforth all be referred to as interviews. We stopped conducting interviews once no new findings were emerging (data saturation was reached). Triangulation was achieved through comparing the findings from the observations and interviews across the different hospitals and through feedback from the HCWs on the main findings of their interviews [35].

We used the Consolidated Framework for Implementation Research [39] to develop the interview topic guide, focusing on its major domains rather than all its elements. Thus although much of our focus was on front-line staff, we explored possible influences on pulse oximeter use from different levels of the health system. Our initial observations helped refine the topic guide. (See S3 Text for the topic guide.) Transcripts were analyzed using thematic content analysis, and the Integrative Model of Behavioural Prediction [40,41] (See Fig 1) was used to frame the findings because this model appeared to provide the best means to organize our explanatory findings. The underpinning rationale for this model is that whether someone performs a behavior is dependent on whether they intend to do it and whether they experience any barriers.

We closely followed the protocol (see S2 Study Protocol), with the exception of (i) we had planned to interview HCWs from 2 hospitals where pulse oximeters were being used and 2 where pulse oximeters were not being used, but because of changing use of pulse oximeters, we conducted the interviews with HCWs in 2 hospitals where pulse oximeters had been used for some time and 2 hospitals where pulse oximeters had recently been introduced; (ii) because the opportunity arose to speak to HCWs at a CIN network meeting, we conducted shorter interviews there with HCWs from 10 hospitals with a range of levels of pulse oximeter use; (iii) interviews were often shorter than we anticipated (30 minutes to 1 hour) because of HCW time constraints and because topics were sufficiently covered in less time; (iv) the topic guide included in the protocol was iteratively refined during the process of conducting the

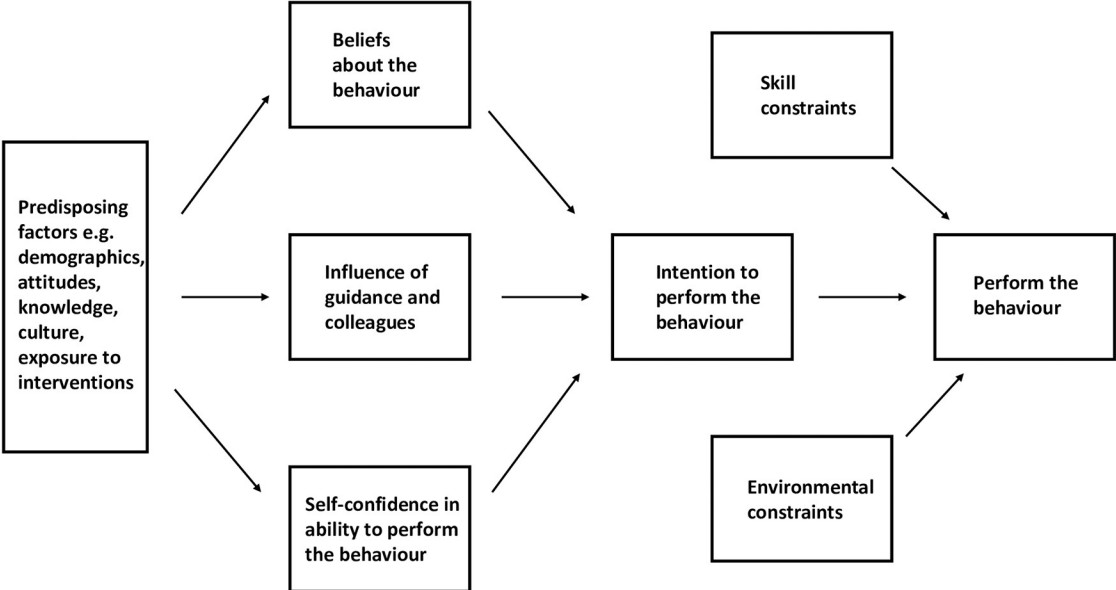

**Fig 1. The Integrative Model of Behavioural Prediction, adapted from Fishbein and Yzer and Fishbein and Ajzen [40,41].**

interviews; and (v) instead of the topic guide being piloted with a doctor or nurse in a hospital where pulse oximeters are used and one where they are not used, a range of HCWs and researchers familiar with the hospitals were consulted.

## Results

We analyzed data that were collected from the 7 CIN hospitals where pulse oximeters had been used with at least 20% of children admitted to the hospital. The hospitals are located in 5 counties, and between 20% to 50% of their catchment population were living in extreme poverty [42]. We also interviewed 30 HCWs from across the 14 CIN hospitals. (See Fig 2 and S1 Table).

### Variability in pulse oximeter use

Analysis of the CIN data set show that from September 2013 through February 2016, 13 of the 14 CIN hospitals had at least 1 pulse oximeter. Pulse oximeters were not used at 2 of the hospitals where they were available and were used with 7% to 75% of children at the remaining hospitals. Pulse oximeter use varied within hospitals over time. Overall, there was some evidence of an increased use followed by a decrease, and then a subsequent further increase. HCWs at 4 hospitals almost never used pulse oximeters, whereas at 3 other hospitals, pulse oximeter uptake began in the last 6 to 12 months of the study period. (See Fig 3).

### Study population

We included 27,906 child admissions from 7 hospitals in the analyses, ranging from 2,700 to 5,700 admissions per hospital. The median age of the children was 1 year, and 55% were male. Three-quarters had a fever, over half had a cough; other symptoms/signs were difficulty breathing (34%), difficulty feeding (34%), and indrawing (32%). Roughly half of these children had a pulse oximeter reading, and of these, 10% had an oxygen saturation level below 90%.

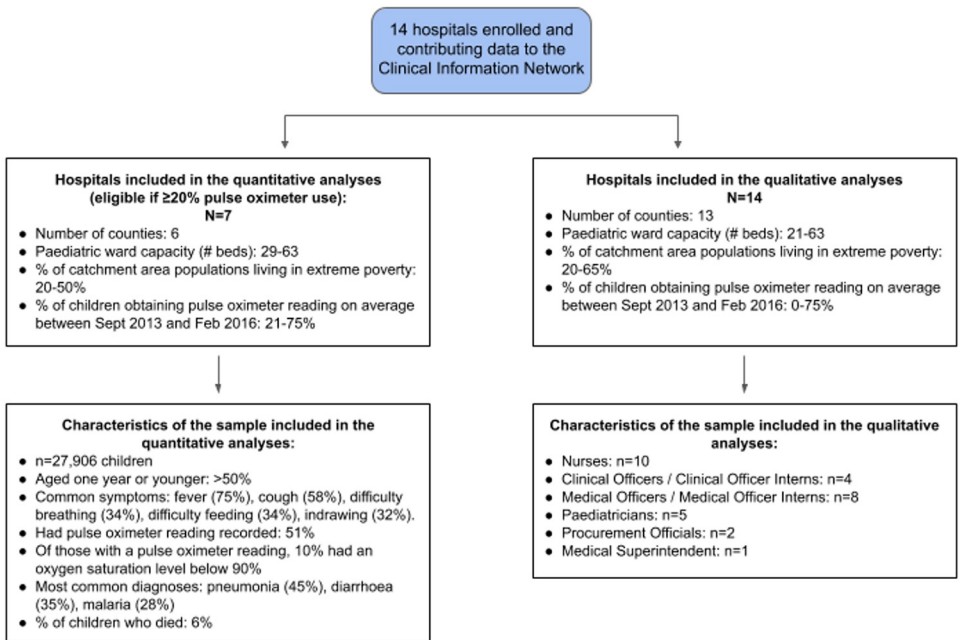

**Fig 2. Flow diagram of the hospitals included in the study and sample characteristics.**

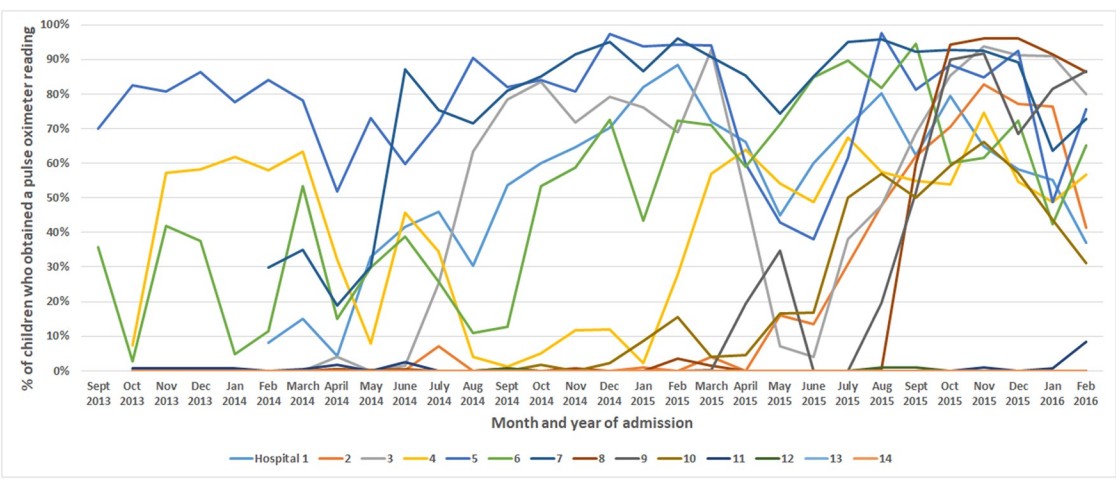

**Fig 3. Pulse oximeter use over time by hospital.**

The most common diagnoses were pneumonia, diarrhea, and malaria: 45%, 35%, and 28% of children, respectively, had these diagnoses. Children often had multiple diagnoses; for instance, of the children with pneumonia, 29% also had diarrhea, 17% had malaria, and 16% had dehydration. Approximately 6% of admitted children died (see S2 Table).

## Children with whom pulse oximeters are used

Results from the multivariable analyses indicate that, when all other variables were held constant, a pulse oximeter was more likely to be used in certain hospitals and the likelihood increased over the study time period. Pulse oximeter use was more likely when an admission checklist (the Paediatric Admission Record [PAR]) was present regardless of whether it was used (OR PAR used compared with PAR not present: 2.41, 95% CI: 1.98, 2.94, $p < 0.001$). The other factors most influential on whether or not a pulse oximeter was used were if the child was not alert (OR: 1.30, 95% CI: 1.09, 1.55, $p = 0.003$), had chest indrawing (OR: 1.28, 95% CI: 1.17, 1.40, $p < 0.001$), or a very high respiratory rate (OR: 1.27, 95% CI: 1.13, 1.43, $p < 0.001$), or if the child was not alert and had indrawing (OR: 0.75, 95% CI: 0.57, 0.98, $p = 0.03$). Pulse oximeters were also more likely to be used with children who were admitted on a weekday; had a fever, cough, difficulty breathing; were not vomiting everything; and/or did not have difficulty drinking. The likelihood of pulse oximeter use decreased with an increase in age (see Fig 4 and S3 Table).

Providing further insight into these findings, all interviewed HCWs intend to use pulse oximeters with at least certain groups of children at admission, if not with all. According to the interview findings, many HCWs consider that pulse oximeters should be used with all children at admission. Others listed various "children who don't need pulse oximeters" (8; clinical officer) and so for whom "doing an SpO2 doesn't really make sense" (3; medical officer). When these HCWs were asked which children do not need a pulse oximeter, the most commonly identified characteristic was those without respiratory symptoms, or more specifically, those with diarrhea, vomiting, or physical trauma.

There is a widespread belief that pulse oximeters are important and useful, "a life-saver" (3; medical officer) and "we are lucky to have it" (16; nurse). HCWs reported that pulse oximeters are particularly useful for guiding decisions on when to give oxygen, for helping to know when a child is improving or deteriorating, and "it can assist you, to titrate (the oxygen) and give the

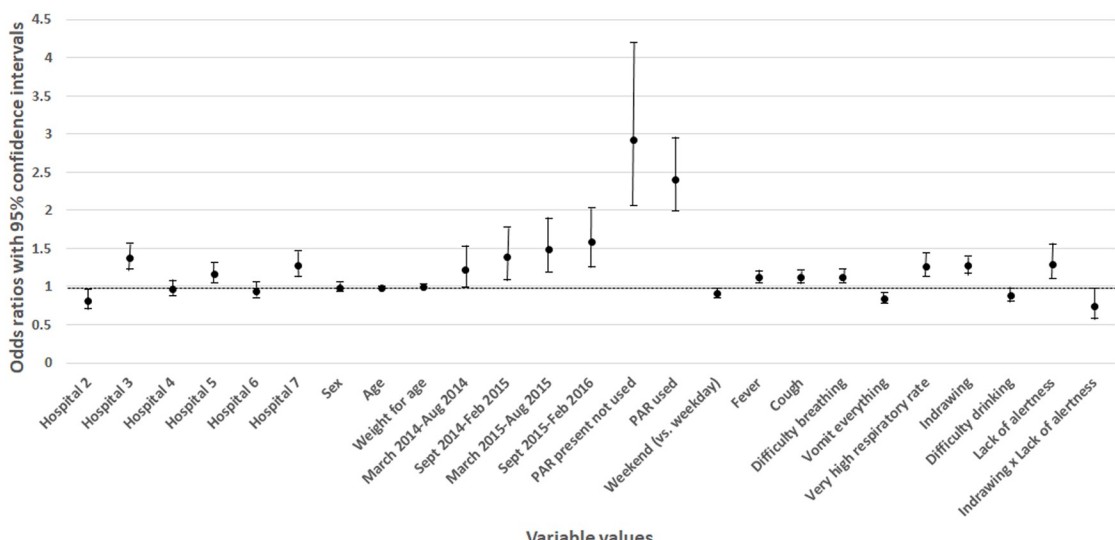

**Fig 4. ORs of the regression model examining with which children pulse oximeters are most likely to be used.** OR, odds ratio; PAR, Paediatric Admission Record.

minimum you need" (2; pediatrician), which is important given that oxygen "is expensive, especially in our setup. We need to really use it cost-effectively" (2; pediatrician).

According to the interviews, HCWs are confident in their ability to use pulse oximeters because they are believed to be very easy to use. If senior consultants/pediatricians insist on pulse oximeter use, this provides strong endorsement. Thus some HCWs explained that their consultants "really insist on the importance" (11; clinical officer) of using pulse oximeters, and "every day they ask, how much was the pulse oximetry" (9; medical officer). The result is that "It's like a chain, yeah; the consultant ask me, and me I ask the intern, how come you didn't do this. So it becomes like a norm or a culture, like everyone has to do $SpO_2$ for every baby" (9; medical officer). "It happens from the top" (8; clinical officer). Guidance was also found to be influential; HCWs frequently mentioned the Kenyan Basic Paediatric Protocols, appeared to trust them, and spoke about an increased awareness of the Protocols amongst the staff.

## Implications of pulse oximeter use

According to our multivariable model, after adjusting for patient level covariables, children with whom pulse oximeters were used had a 42% higher odds of being prescribed oxygen than children with whom pulse oximeters were not used (OR: 1.42, 95% CI: 1.25, 1.62, $p < 0.001$). The healthcare factors that were most associated with the use of oxygen therapy were hospital and admission time period; the symptoms associated with oxygen therapy were chest indrawing (OR: 2.69, 95% CI: 2.33, 3.09, $p < 0.001$), difficulty breathing (OR: 2.00, 95% CI: 1.76, 2.28, $p < 0.001$), and cyanosis (OR: 1.92, 95% CI: 1.34, 2.76, $p < 0.001$); and the diagnoses were asthma (OR: 1.94, 95% CI: 1.59, 2.35, $p < 0.001$), bronchiolitis (OR: 1.88, 95% CI: 1.47, 2.41, $p < 0.001$), and pneumonia (OR: 1.76, CI: 1.49, 2.07, $p < 0.001$; see S4 Table and S1 Fig).

After controlling for all other factors, including patient level covariables, the value of the pulse oximeter reading was a significant predictor of oxygen provision. The odds of a child being prescribed oxygen was more than twice as high if they had a pulse oximeter reading below 90% than if they had a pulse oximeter reading of 90% or more (OR: 3.29, 95% CI: 2.82, 3.84, $p < 0.001$).

Complementing these quantitative results, many interviewees independently identified the 90% threshold that is recommended in the Kenyan Protocols as the one they use in daily practice. HCWs indicated that when they use pulse oximeters at admission, they predominantly use them to decide whether to give children oxygen or to provide a baseline for monitoring. However, the findings suggest that clinical signs are often prioritized over pulse oximeter results when there is uncertainty about interpreting a low pulse oximeter reading that contradicts clinic signs. In these cases, HCWs may presume the pulse oximeter is malfunctioning and follow the indications of the clinical signs, either providing or withholding oxygen. Pediatricians and medical officers were more likely than clinical officers (nonphysician clinicians) to describe situations when they were concerned that low pulse oximeter results contradicted clinical signs, and nurses were least likely. Although sometimes the pulse oximeter is malfunctioning (because it is faulty or because it is being handled incorrectly), in some cases, the HCW may not be aware that children can have hypoxemia without exhibiting severe respiratory symptoms.

> "In some ways they are useful, in others they aren't. Because, a child can come, they are completely stable. No difficulty in breathing, respiratory rate is ok, but when you put the SpO2 machine, it shows you the child is at 80%, but when you look at the child clinical, ah you don't think they should be on oxygen. So that's one let down."
>
> (12; medical officer)

> "If the device was dropped down maybe it's faulty because of that, so mechanical damage can give you false reading, other times, I don't know, poor quality pulse oximeters, they just give you funny readings."
>
> (3; medical officer)

> "I wouldn't really point where the problem is; is it about, we operating the machine, or is it that the machine is broken, or spoilt."
>
> (10; nurse)

## Barriers to pulse oximeter use

Almost all of the interviewed HCWs have experienced barriers to pulse oximeter use, often frequently. Inadequate availability was arguably the main barrier to pulse oximeter use discussed by the HCWs:

> "There are situations when you want to use it (a pulse oximeter) but you can't use it, due to things you can't control."
>
> (3; medical officer)

> "The pulse oximeters here are very useful. The problem is they are very few.(. . .) We always need, it's, there's always one person who needs it but you're using it, so it happens every time, (. . .) every day."
>
> (8; clinical officer)

Lack of availability is due to a perceived lack of prioritization of pulse oximeters by the hospital and/or county, a complicated and unclear procurement process, and unavailability of

ward pulse oximeters for long periods of time once they break because of inefficient repair processes.

> "It can take months to years to get new equipment after a requisition for it."

(2; pediatrician)

> "You have to talk to the finance department, the county have to actually assess or to give approval of funds to be available for them to be repaired. It's quite a long process. Yeah. It's not something that it would be spoilt this week and next week it's done; it can even take months before you get another one."

(9; medical officer)

HCWs recognize that a key reason for hospitals not providing enough pulse oximeters is that they have limited resources to buy new equipment. However, the procurement process itself was more widely criticized for taking a long time and lacking transparency.

Other reasons for repair delays include this explanation given by a pediatrician:

> "I can say it's a problem with the Biomedical department maybe, because if you have an audit of the equipment you have, and you should be doing maintenance, you should have an idea what the spare parts that are needed, and maybe procure them in time, or have a way of repairing them as soon as they break down. So it's either the, they're not up to date, or competent enough, or their training also has some issue maybe, because sometimes they will not be able to do anything, or they can't tell you what has happened, so you don't know who is going to help you."

(2; pediatrician)

Other common environmental constraints include batteries running out and taking a long time to be charged or replaced and not having appropriately sized probes.

> "Most of them don't come with probes for a neonate, for a bigger child, so(. . .) sometimes we really struggle to try and get an SpO2, because the machine we have only has a probe that is big, for a big finger."

(2; pediatrician)

An insufficient oxygen supply was reported by some of the hospitals. However, HCWs often feel the more substantial barrier to giving oxygen to all children who need it is an inadequate availability of fully functioning flow meters and splitters (to measure how much oxygen is given, and so HCWs can attach multiple children to one oxygen cylinder/concentrator, respectively), either because of malfunctioning and a lack of repairs or because of inadequate supply.

The main individual level barrier to appropriate pulse oximeter use is because of skill constraints: a lack of training on why and how to use pulse oximeters and how to interpret their results. Thus HCWs sometimes do not have sufficient knowledge of pulse oximeters to use them appropriately.

> "There are people who will not really place it correctly and it gives you a different reading, or it doesn't read at all"

(1; medical officer)

When asked how HCWs within their ward could be encouraged to use pulse oximeters more often with children at admission, HCWs most commonly recommended obtaining more pulse oximeters and providing training on how and why to use pulse oximeters.

"Why should we take SpO2: if you don't know, are you going to put a lot? You're not going to put a lot of effort. But if you know the reason as to why you are doing this, you'll go ahead and wish to know."

(15; nurse)

"They just need to be sensitized, that you have not completed admitting a patient, a child, until you have done a pulse oximetry reading"

(2; pediatrician)

"A mindset change [is necessary]."

(8; clinical officer)

In summary, HCWs sometimes do not use pulse oximeters because of issues with availability or delayed repairs (reflecting wider problems in the Kenyan health system), or because of HCWs' lack of knowledge about why, when, and how to use them.

### Synthesis of quantitative and qualitative findings using the Integrative Model of Behavioural Prediction

In addition to understanding the factors that are associated with whether a HCW chooses to or is able to use a pulse oximeter, the interview findings indicate that whether the pulse oximeter is used correctly, how its reading is interpreted, and the HCWs' resulting follow up actions are also crucial. We adapted the Integrative Model of Behavioural Prediction to include these elements. Insufficient skills in using pulse oximeters and interpreting their results correctly are influential, as are environmental constraints in the form of supply and repair of pulse oximeters and available oxygen therapy equipment. These barriers can limit appropriate pulse oximeter use and explain variation in use both between and within hospitals over time. Fig 5 shows the adapted model with each component identified from the data.

### Discussion

In this analysis of a large data set established by a CIN in Kenya and interviews with hospital staff, we assessed how a low-cost and effective intervention—pulse oximetry—was implemented within the Kenyan health system. In addition to specific clinical signs, a simple PAR, the hospital and the time period of admission predicted the use of pulse oximeters in children admitted to Kenyan hospitals, and children with a pulse oximeter reading were more likely to be prescribed oxygen. Findings indicate that an inadequate supply, broken pulse oximeters, and insufficient training are the main barriers to pulse oximeter use; comparable findings have been reported by others [15,16,21,22,43–46]. Our findings highlight that improving the supply of reliable pulse oximeters is crucial, as is ensuring a consistent supply of oxygen and associated equipment such as flow meters. If possible HCWs should have the option to request the pulse oximeter type that best suits their clinical situation, for example, large stand-alone monitors versus small, portable forms, and be supplied with probes of various sizes for newborns, infants, young children, and older children.

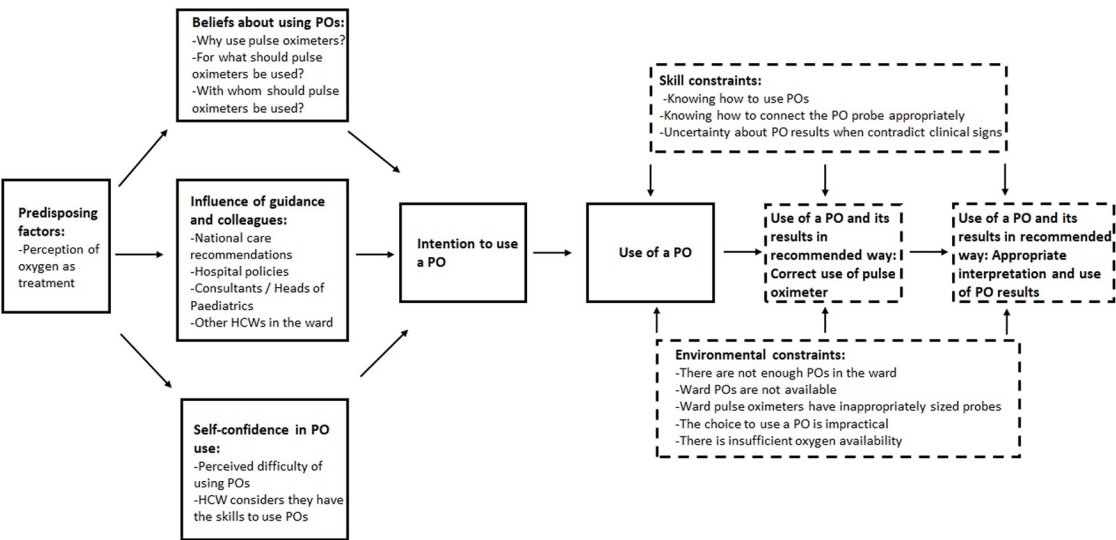

**Fig 5. Adapted Integrative Model of Behavioural Prediction.** Dashes indicate components of the model that we modified or added to reflect the findings. HCW, healthcare worker; PO, pulse oximeter.

Variation between hospitals in the use of pulse oximeters also appears to be due to the leadership role played by senior doctors who advocate for the importance of pulse oximetry [21,47,48], county officials who value the importance of pulse oximeters, and/or strong relationships between the hospitals and donors who provide pulse oximeters. In those hospitals where pulse oximetry was rarely or never used, the main reason discussed was a lack of access to pulse oximeters. HCWs from these hospitals reported that they understand the importance of pulse oximeters and want to use them but do not have the social or political capital to ensure this type of equipment is procured. It is worth noting that in Kenya, unlike with drugs, no central process exists to support the purchase of medical equipment such as a pulse oximeter.

The variability within hospitals over time (see Fig 3) is likely due to delays in pulse oximeter repairs. Pulse oximeter use increased over time, and this was probably due to the feedback provided by the CIN to hospitals [26]. This feedback focused on the completion of documentation and adherence to guidance that supports the use of pulse oximetry [24,33]. In the interviews, HCWs reported that CIN reports and meetings educate and motivate them to improve care and compete with other hospitals [47]. The dissemination and use of the Kenyan Basic Paediatric Protocols also likely supported the increasing use of pulse oximeters [23].

The PAR was designed to improve the recording of clinical information and to act as a prompt for recommended clinical procedures [49]. The data set analyses indicate that availability of the PAR was associated with increased pulse oximeter use, and interview participants explained that the PAR reminds them to use a pulse oximeter with each child and underscores the importance of doing so. Clinical reminders have been used in other settings, with mixed results [50–57]. However, the PAR is a specific type of reminder because it is a checklist that is integrated into routine workflows rather than an additional form to be completed.

When HCWs use pulse oximeters at admission, they predominantly use them to decide whether to give children oxygen or as a baseline measure for monitoring. The children most likely to be prescribed oxygen as a result of a low pulse oximeter reading is largely consistent with the oxygen provision guidelines of the WHO's 2016 Oxygen Therapy for Children manual [58] and the Kenyan Basic Paediatric Protocols. There is some uncertainty in providing oxygen to children with low pulse oximeter readings if their clinical signs appear to contradict

the pulse oximeter reading; this could lead to a child not receiving oxygen when required. Having difficulty dealing with this conflicting information, and the perception that pulse oximeters can be unreliable, might be due to previous experience of pulse oximeters malfunctioning. This is more likely when pulse oximeters are few and therefore overused and when HCWs have to use inappropriately sized probes. Another contributing factor is HCWs' uncertainty that children can require oxygen without exhibiting clinical signs and about situations in which pulse oximeters may be less accurate [59–64]. Furthermore, as interviewees discussed, sometimes HCWs use pulse oximeters incorrectly because of a lack of training, and this could lead to inaccurate readings. Training staff to recognize which groups of children might benefit from pulse oximeter readings, and why, has been noted elsewhere [16,65–67] and has been reported to be effective in Nigerian hospitals [21].

Pediatricians and medical officers were more likely than clinical officers to describe situations when they were concerned that low pulse oximeter results contradicted clinical signs, and nurses were least likely. This is consistent with other studies of guideline adherence in which the most senior practitioners may be least adherent [22,68,69]. However, in general, similar opinions and experiences about pulse oximeter use were expressed and discussed across the different hospitals and groups of HCWs, even though CIN hospitals vary in size, number of admissions, location, socioeconomic profile, and pulse oximeter use.

Interviews indicated that there is an inconsistent use of the 90% threshold, which mirrors the debate continuing in the literature on whether 90% is the threshold that should be used. Furthermore, several studies have reported that allowing lower oxygen saturation levels (termed "permissive hypoxemia"), particularly in children with less severe disease, might not be harmful [70]. Further research is required on the impact of using the 90% threshold or alternative thresholds on HCW decision-making, health outcomes, and resource utilisation [71] and on the benefits and harms of treatment with or referral for oxygen in children who do not appear to be clinically severely ill.

The CIN provides a rare example of a large clinical data set in a LMIC setting that can be used to assess how available low-cost effective interventions are used in practice. The data are relatively high quality because of regular monitoring and periodic feedback to hospitals on the quality of data collection [24]. Through a sequential approach, we used the interview findings to validate and interpret the results of the analysis of the CIN data set. This enabled us to examine topics that the quantitative methods alone could not explore, for example, HCW beliefs and barriers to using pulse oximetry. We adapted the Integrative Model of Behavioural Prediction to assess whether pulse oximeters are used appropriately in addition to being used. This extended model may be useful to others examining health technology adoption. However, we acknowledge that a number of other models and frameworks could have been used (for example, those of Damschroder and colleagues, Michie and colleagues, and May and colleagues [39,72,73]).

One potential limitation of this analysis is that we included children with high levels of missing data. Although the multiple imputation process might be less accurate for these individuals, we felt these children would contribute valuable data and including them would maximize the data available for analysis. The risk of bias from including these data was low because of the large number of children contributing to the analysis; also, few children had very high missing data levels. Furthermore, we conducted sensitivity analyses that did not find any violation of the MAR assumption (see S1 Text). A second limitation is that the HCWs who consented to participate in the interviews might not have been representative; for example, those who agreed might have been less busy and thus had different patterns of pulse oximeter use, opinions, and experiences (nonresponse bias [74]). Our interviews did not enable detailed exploration in an explanatory sense of differences between counties, or of senior management

differences across counties and hospitals, but focused on common features. This limits our ability to draw conclusions on differences between locations. Furthermore, we did not record the types of pulse oximeters used in the wards where interviews were conducted, so findings cannot be mapped to pulse oximeter type.

## Conclusion

We examined the use of pulse oximeters and barriers to their use in the LMIC healthcare context of Kenyan county hospitals. We found that pulse oximeter use varies substantially between and within Kenyan hospitals over time. HCWs were most likely to use pulse oximeters with children with a very high respiratory rate, indrawing, and/or who were not alert; children who obtained a pulse oximeter reading were more likely to be prescribed oxygen than if a pulse oximeter was not used; and children with a reading below 90% were more likely to be prescribed oxygen than those with higher readings, suggesting that HCW decision-making is influenced by international and national guidelines. However, HCWs cannot always use pulse oximeters when they intend to because of a lack of availability, complex procurement processes, and repair delays. Furthermore, HCWs sometimes use pulse oximeters incorrectly or misinterpret their results because of insufficient training.

We can conclude from our findings that it is important for hospitals to have efficient and transparent procurement, maintenance, and repair systems to ensure continuous adequate availability of pulse oximeters; meaningful leadership from senior doctors advocating for pulse oximeter use; further training for HCWs of all cadres and experience levels on how, when, and why to use pulse oximeters; and a system for monitoring and receiving feedback on recommended practice.

Our findings can inform the design of programs to increase the use of pulse oximeters with children at admission to a hospital and thereby increase the chance of prompt and effective treatment, in line with the Sustainable Development Goals. Furthermore, in supporting effective treatment delivery, increased pulse oximeter use might also lead to hospital cost savings by ensuring oxygen is targeted at those who are most likely to benefit. The potential for health outcome improvement in children with pneumonia is particularly encouraging given that half of the children in the hospitals we studied have pneumonia, which is also the leading infectious cause of death in children aged 1 month to 5 years worldwide.

## Supporting information

**S1 Table. Characteristics of the CIN hospitals from which quantitative and qualitative data were collected.** CIN, Clinical Information Network.
(DOCX)

**S2 Table. Descriptive statistics of the children admitted to the 7 CIN hospitals during the study period.** CIN, Clinical Information Network.
(DOCX)

**S3 Table. ORs and CIs produced from the logistic regression investigating with which children pulse oximeters are used.** OR, odds ratio.
(DOCX)

**S4 Table. ORs and CIs produced from the logistic regression investigating the factors influencing whether oxygen is prescribed.** OR, odds ratio.
(DOCX)

**S1 Text. Determining whether the data could be considered MAR.** MAR, Missing at Random.
(DOCX)

**S2 Text. Developing a best-fit regression model.**
(DOCX)

**S3 Text. Semistructured interview topic guide.**
(DOCX)

**S1 Study Protocol. Quantitative research protocol.**
(DOCX)

**S2 Study Protocol. Qualitative research protocol.**
(DOCX)

**S1 Fig. ORs of the regression model examining factors affecting oxygen provision.** OR,
odds ratio.
(TIF)

**S1 STROBE Checklist. STROBE checklist.** STROBE, Strengthening the Reporting of Observational Studies in Epidemiology.
(DOCX)

## Acknowledgments

We would like to thank The Kenyan CIN, including Samuel N'gar n'gar (Vihiga County Hospital), Nick Aduro (Kakamega County Hospital), David Kimutai (Mbagathi County Hospital), Cecilia Mutiso and Celia Muturi (Mama Lucy Kibaki County Hospital), Charles Nzioki (Machakos County Hospital), Agnes Mithamo (Nyeri County Hospital), Magdalene Kuria (Kisumu East County Hospital), Samuel Otido (Embu County Hospital), Peris Njiiri (Kerugoya County Hospital), Rachel Inginia (Kitale County Hospital), Barnabas Kigen (Busia County Hospital), Lydia Thuranira (Kiambu County Hospital); colleagues from the KEMRI-Wellcome Trust Research Programme: Jacquie Oliwa, Jacinta Nzinga, Grace Irimu, David Gathara, Sam Akech, Morris Ogero, Mercy Chepkirui, and George Mbevi; Orlaith Burke and Sofia Massa (Nuffield Department of Population Health, University of Oxford); and all of the interview participants.

## Author Contributions

**Conceptualization:** Abigail J. Enoch, Mike English, Sasha Shepperd.

**Data curation:** Mike English.

**Formal analysis:** Abigail J. Enoch.

**Investigation:** Abigail J. Enoch.

**Methodology:** Abigail J. Enoch, Mike English, Gerald McGivern, Sasha Shepperd.

**Supervision:** Mike English, Sasha Shepperd.

**Visualization:** Abigail J. Enoch.

**Writing – original draft:** Abigail J. Enoch.

**Writing – review & editing:** Abigail J. Enoch, Mike English, Gerald McGivern, Sasha
Shepperd.

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
