## [Decision Letter · Decision Letter 0]

23 Sep 2019

Dear Dr. Enoch,

Thank you very much for submitting your manuscript "Variability in the use of pulse oximeters with children in Kenyan hospitals: a mixed-methods analysis" (PMEDICINE-D-19-02955) for consideration at PLOS Medicine. 

[LINK]

In light of these reviews, I am afraid that we will not be able to accept the manuscript for publication in the journal in its current form, but we would like to consider a revised version that addresses the reviewers' and editors' comments. Obviously we cannot make any decision about publication until we have seen the revised manuscript and your response, and we plan to seek re-review by one or more of the reviewers. 

We expect to receive your revised manuscript by Oct 07 2019 11:59PM. Please email us (plosmedicine@plos.org) if you have any questions or concerns.

We look forward to receiving your revised manuscript. 

Sincerely,

Clare Stone, Editor-in-Chief, 

for 

Louise Gaynor, MBBS PhD

Associate Editor 

PLOS Medicine

plosmedicine.org

Comments from the reviewers:

Reviewer #1: Overall this is an important paper on a key topic. I feel with some amendments this could be an excellent addition to the learnings around the introduction of pulse oximetry in these settings. Overall I think there needs to be further work done on clearly outlining the methods, results and conclusions in a more precise way. There is inconsistency in how the results are presented in the abstract, findings and conclusions and this can be confusing.

Specific comments:

- Abstract methods - framework guided your design and then you conducted quan and qual data collection to answer specific outcomes - please describe this more clearly

- Abstract findings - inconsistent with findings and conclusions (conclusion seems the clearest so perhaps use this)

- Abstract conclusions - first sentence is not a conclusion

Page 4 L72 infectious cause of disease

P4 L82 - include other references here - such as H Graham and King et al

P5 L112 - add more about the settings, hospitals included and rationale for this

P6 L125 onwards - need to much more clearly explain the methods used and specifically describe the variables used for quan analysis (perhaps list in appendix) and also the qual methods - you mention "direct (non-participant) observations" - what are these and what data did you collect and how did you analysis it and where is it presented and discussed? Also you mention "semi-structured conversations" but you did have topic guides so they are more than conversations?

P10 - Super interesting data but perhaps you can find a better way to present this

P11 - Study population - great data here - would be nice to have a flow to show this data and the outcomes etc. - 45% pneumonia is extremely high and should have been discussed more - especially since you only saw 10% hypoxemia?

P11 L216 onwards I am not sure this is the clearest way to present results - I would keep quant and qual separate and bring them together in the discussion - also list OR in descending order and again look at diagram and perhaps explain why you have selected these variables from all 45?

P14 Use more headings and not sure how you are referencing qualitative data - I would use longer quotes as it is hard to understand the data as presented

P19 L373 how the individual uses the pulse oximeter , recognising...

P20 Great diagram - I would show the original up front in the methods and then show the results one here 

P21 Discussion - again please check this relates to the data being presented - e.g. L385 the main barriers - is this presented in the data earlier? L394 'leadership role played by senior doctors' again would be good to present this in the results first and have some quotes

P22 L414 Does the data not also show something about adherence to guidelines? Would be interesting to discuss this and how it seems they are following iMCI and improving over time? Lots of data showing this is not always the case

P25 Strengths and weaknesses - L496 is this not a limitation as presented now? and the same in P26 L511?

P26/27 Check conclusions against data and align them

Reviewer #2: Alex McConnachie, Statistical Review

Enoch and colleagues report a mixed methods study of the use of pulse oximeters in Kenyan hospitals with children, utilising quantitative analysis of routine data, and qualitative analysis of hospital staff interviews. This review is focused on the quantitative side, specifically the use of statistical methods.

As a whole, I thought the paper reads quite well, and the statistical methods used in the paper were good. My comments are generally to do with the presentation of the results.

In the abstract, we are told that a PO was more likely to be used "…in certain hospitals, at later time periods, when a Paediatric Admissions Record was used, and if they had a very high respiratory rate (OR: 1.27, 95% CI: 1.13, 1.43)…" It is not clear why ORs for the first 3 associations are not reported, but they are from then on.

The next paragraph of the abstract then starts to talk about the main barriers to PO use, but it is not clear at that point where these results are coming from. Whilst it becomes clearer after reading more of the paper that these are qualitative findings, at the point of reading the abstract, this was not so obvious.

In the methods, lines 157-158, it states that missing data patterns were explored and the data could be treated as MAR. I often see statements like this, but have yet to find anyone who can explain how this judgement can be made. What was it in the pattern of missing data that was observed that led to this conclusion? Also, the discussion (around line 515) mentions sensitivity analyses that are not described in the methods section, which purportedly tested the MAR assumption. It is not clear what these sensitivity analyses were. 

Multiple imputation for missing data is good to see, but I thought the "M" in MICE stood for "Multiple", not "Multivariate".

The last paragraph of the (quantitative) methods section mentions checking the robustness of the results using a bootstrapping method. I can imagine the type of analysis this might entail, but again, there is not enough detail. I wonder whether it would be worth including some more information about these methods (and others noted above) and some results, perhaps in a supplement?

Figure 2 looks quite nice, but is not the best way to show these results. A table would be better, as the magnitude of the odds ratio may not give an accurate impression of the importance of each predictor. Also, there is no table showing the characteristics of the population in relation to all of the factors considered, and summarising the rate of PO use within subgroups - it is important to show the actual data (including missing data levels), and to report the univariate associations with PO use. Showing all of the potential predictors is important so that we can see which predictors made it into the final model, and which did not.

The same goes for predictors of supplemental oxygen use; though here we are not even given a figure to summarise the final model. One result mentioned in the paper was that those with a PO reading below 90% were more likely to receive oxygen, with the conclusion being that HCWs were influenced by the recommended 90% threshold. This seems speculative - there could simply be a trend with more oxygen use with lower PO readings (i.e. there may be nothing in particular about the 90% threshold), but this could be investigated to see if the rate of oxygen use changes sharply around the 90% threshold.

Reviewer #3: Thank you for the opportunity to review this manuscript. The article addresses and important question in the context of existing literature on oximeter use in LMIC settings and appropriately asks questions about oximeter use - when, with which children, with what outcomes? It offers practical suggestions and implications of the work done.

Introduction 

Well written and covers the scope of issues and appropriate literature.

Methods

This is a well written paper which appropriately uses a sequential mixed methods design to answer questions regarding how, when and why of oximeter use.

The rationale for selection of 7 hospitals is well described.

It would be good to tighten the description of the qualitative data sources. It seems there is quant data from 7 hospitals - but qual data from at least 14 hospitals (as described in the abstract)but less clear in the results section itself) plus observation in some. Clearly qual data came form hospitals from which quant data was also collected but it difficult to get a clear pattern of the data sources. Could this be represented in a tabular or other format for clarity? Although the use of more than one source of qual data (eg interviews and observation) increases robustness through triangulation it is not clear in the methods or the results how the observational data was used or triangulation was used. It would also be useful to explain how the 3 observed hospitals were chosen eg) convenience vs particular characteristics to help understand the context of the data as this may change how this data should be considered in the overall picture.

Results and discussion

It would be good to increase the clarity or tighten the connection between the text description of the qualitative data and the labels used in the synthesis of the data in Figure 3 to make it easier to understand which parts of the text description relate to which part of this framework. 

Some of this information is raised in the discussion eg) influence of protocols and guidelines and feedback from CIN data motivating improvement. If this came from the interviews - should it be part of the results rather than in the discussion (I may have missed it an apologise if I did)? 

Discussion

Suggested edits for understanding/ease of reading/language are as follows

Line 44 Suggest rewording in findings: "...children with a reading below 90% were more than twice as likely to RECEIVE oxygen....

Line 150 suggest "different health issues and treatments compared to older children"

Line 216 For which children pulse oximeters are used - otherwise use a ? at the end of this theme heading.

Line 225 I am not sure children "obtain" a pulse oximeter reading "have a pulse oximetry performed" "have a SpO2 recorded" 

Line 234 onwards - suggest italics for quotes to make it clearer what is quote and text

Line 234 to 238 is a long and sometimes difficult to read sentence. Suggest breaking into more than one.

Line 254 "given a pulse oximeter reading" - the language around this needs to be tightened as per above.

Line 263 "obtain oxygen" suggest "receive oxygen"

Line 265-6 I wonder if the sentence "suggesting HCWs are influenced by the recommended 90% threshold is more appropriate for discussion where it is already elaborated on. Alternatively this could be explicitly stated in Figure 3 in Influence of guidelines ie Oxygen for SpO2<90% (rather than referring generically to "guidelines")

Line 273-274 suggest - follow clinical indications, either providing or withholding oxygen.

Line 363 In summary,...

Please review sentence length throughout- sentences with a lot of brackets, colons and qualification sometimes lose their impact through their length

Figure 3; In the box "self confidence in PD use: the second dot point needs revision for grammar

[LINK]

---

## [Editor Report · Decision Letter 1]

24 Oct 2019

Dear Dr. Enoch,

Thank you very much for re-submitting your manuscript "Variability in the use of pulse oximeters with children in Kenyan hospitals: a mixed-methods analysis" (PMEDICINE-D-19-02955R1) for review by PLOS Medicine.

I have discussed the paper with my colleagues and the academic editor . I am pleased to say that provided the remaining editorial and production issues are dealt with we are planning to accept the paper for publication in the journal.

[LINK]

We look forward to receiving the revised manuscript by Oct 31 2019 11:59PM. 

Sincerely,

Louise Gaynor, MBBS PhD

Associate Editor 

PLOS Medicine

plosmedicine.org

Requests from Editors:

Non-author data contact or URL details needed for reserchers who meet criteria for access to data -PLOS Medicine requires that the de-identified data underlying the specific results in a published article be made available, without restrictions on access, in a public repository or as Supporting Information at the time of article publication, provided it is legal and ethical to do so. Please see the policy at 

http://journals.plos.org/plosmedicine/s/data-availability

 I don't think the mention of "mice" (abstract) is needed here (main text is fine); on the other hand, please include summary demographic and clinical details on the children; As the final sentence of the methods and findings section of the abstract please add a sentence in the limitations…starting ‘ limitations of the study are…’

Please add p values throughout, where 95%Cis are given

Please ensure any questionnaires (you mention interviews) are provided as Supp files. 

At this stage, we ask that you include a short, non-technical Author Summary of your research to make findings accessible to a wide audience that includes both scientists and non-scientists. The Author Summary should immediately follow the Abstract in your revised manuscript. This text is subject to editorial change and should be distinct from the scientific abstract. Please

see our author guidelines for more information: https://journals.plos.org/plosmedicine/s/revising-your-manuscript#

Did your study have a prospective protocol or analysis plan? Please state this (either way) early in the Methods section.

Square brackets for refs in main text – Please place before the full stop at the end of the sentence. Multiple refs should be all within one set of brackets, line 442 as an example, with ref numbers separated by commas. 

Some idiosyncrasies in the references (e.g. absent full access details for the Cochrane reviews 51-53). 

Please provide a STROBE checklist as a supp file and please use sections and paragraphs as these can change on publication.

Was written consent provided for children?

Comments from Reviewers:

[LINK]

---

## [Editor Report · Decision Letter 2]

21 Nov 2019

Dear Dr Enoch, 

On behalf of my colleagues and the academic editor, Dr. James K Tumwine, I am delighted to inform you that your manuscript entitled "Variability in the use of pulse oximeters with children in Kenyan hospitals: a mixed-methods analysis" (PMEDICINE-D-19-02955R2) has been accepted for publication in PLOS Medicine. 

PRODUCTION PROCESS

PRESS

PROFILE INFORMATION

Thank you again for submitting the manuscript to PLOS Medicine. We look forward to publishing it. 

Best wishes, 

Louise Gaynor-Brook, MBBS PhD

Associate Editor 

PLOS Medicine

plosmedicine.org